# Proton Pump Inhibitors Worsen Colorectal Cancer Outcomes in Patients Treated with Bevacizumab

**DOI:** 10.3390/cancers16193378

**Published:** 2024-10-02

**Authors:** Chin-Chia Wu, Chuan-Yin Fang, Wen-Yen Chiou, Pei-Tsen Chen, Ta-Wen Hsu, Shih-Kai Hung, Yu-Tso Liao, Chuan-Sheng Hung, Jui-Hsiu Tsai

**Affiliations:** 1Division of Colorectal Surgery, Dalin Tzu Chi Hospital, Buddhist Tzu Chi Medical Foundation, Chiayi 622, Taiwan; wccstillthought@gmail.com (C.-C.W.); katemole1130@gmail.com (P.-T.C.); daven88888@gmail.com (T.-W.H.); 2School of Post-Baccalaureate Chinese Medicine, Tzu Chi University, Hualien 970, Taiwan; 3Division of Colon and Rectal Surgery, Ditmanson Medical Foundation Chia-Yi Christian Hospital, Chiayi 600, Taiwan; 04969@cych.org.tw; 4School of Medicine, Tzu Chi University, Hualien 970, Taiwan; cwyncku@gmail.com (W.-Y.C.); oncology158@yahoo.com.tw (S.-K.H.); 5Department of Radiation Oncology, Dalin Tzu Chi Hospital, Buddhist Tzu Chi Medical Foundation, Chiayi 622, Taiwan; 6Department of Medical Research, Dalin Tzu Chi Hospital, Buddhist Tzu Chi Medical Foundation, Chiayi 622, Taiwan; 7Division of Colorectal Surgery, Department of Surgery, National Taiwan University Hospital, Hsin-Chu Branch, Hsinchu 300, Taiwan; yutsoliao@gmail.com; 8Department of Computer Science and Engineering, National Sun Yat-Sen University, Kaohsiung 804, Taiwan; scps851210@gmail.com; 9Department of Psychiatry, Dalin Tzu Chi Hospital, Buddhist Tzu Chi Medical Foundation, Chiayi 622, Taiwan

**Keywords:** metastatic colorectal cancer, proton pump inhibitor, H2 receptor antagonists, bevacizumab

## Abstract

**Simple Summary:**

Proton pump inhibitors (PPIs) and H2 receptor antagonists (H2RAs) are commonly used for acid reduction in patients with peptic ulcer disease or esophageal reflux. Previous studies have shown that acid-reducing agents (ARAs) might affect bevacizumab treatment. This study found that PPI use worsened the oncological outcomes in patients with metastatic colorectal cancer (mCRC) undergoing bevacizumab treatment. There was a negative dose–response relationship when compared to H2RA use. Physicians should consider the benefits and risks of long-term use of ARAs, especially PPIs, in mCRC patients treated with bevacizumab.

**Abstract:**

Background: Approximately one-third of patients with advanced colorectal cancer (CRC) and treated with bevacizumab are prescribed proton pump inhibitors (PPIs) or H2 receptor antagonists (H2RAs). However, there is limited data on the effects of PPIs and H2RAs in these patients. To investigate the oncological outcomes of PPI and H2RA use in CRC patients treated with bevacizumab, we performed a retrospective cohort study using the Taiwan National Health Insurance Research Database and Taiwan Cancer Registry Database from 2005 to 2020. Methods: In CRC patients treated with bevacizumab, the PPI users and H2RA users were matched with patients without acid-reducing agents (ARAs) by 1:4 propensity score matching. PPI users and H2RA users were matched with propensity scoring in a 1:1 ratio. We divided patients into 4 cumulative PPI dosage levels to assess the dose–response relationship. The primary endpoints were 5-year overall survival and cancer-specific survival. Results: Compared with ARA non-users, both H2RA users and PPI users were associated with reduced overall survival. PPI users were associated with more significant negative effects on overall survival. Compared with H2RA users, PPI users were associated with lower 5-year overall survival (aHR: 1.19, 95% CI: 1.09–1.31) and cancer-specific survival (aHR: 1.20, 95% CI: 1.09–1.31). A similar dose–response relationship was observed for PPI users in terms of 5-year overall survival and cancer-specific overall survival. Conclusions: Compared to H2AR use, PPI use was associated with dose-dependent poorer oncological outcomes in metastatic CRC patients treated with bevacizumab.

## 1. Introduction

Colorectal cancer (CRC) is one of the most common cancers worldwide and the fourth leading cause of cancer death [1]. About 20% of CRC patients are metastatic at initial diagnosis and another 25% patients will go on to develop metastases [2]. The inhibitor of vascular endothelial growth factor (VEGF) is one of the important target therapies used for treatment of metastatic cancers, including colorectal, breast, ovarian, and non-small cell lung cancers [3,4]. Bevacizumab is a recombinant humanized monoclonal antibody that selectively binds to circulating VEGF. Combined chemotherapy with anti-VEGF therapy is approved for use to prolong overall and progression-free survival in metastatic CRC (mCRC) patients [5].

While anti-VEGF treatment provides numerous benefits, there are several side effects as well, such as thromboembolism, gastrointestinal perforation and fistula, hypertension, bleeding, and allergies [6]. Interactions between bevacizumab and other drugs were discussed and reported previously. Peptic ulcer, reflux esophagitis, and stress ulcer are common in cancer treatment and reported in approximately 20–33% of all cancer patients [7]. Gastrointestinal bleeding risk is also increased by bevacizumab [8,9]. Proton pump inhibitors (PPIs) and H2 receptor antagonists (H2RAs) inhibit gastric acid secretion and have been used for the treatment of acid-related diseases [10]. However, there is limited literature about the different effects of PPIs and H2RAs in the advanced CRC patients treated with bevacizumab.

A retrospective analysis found a prevalence of acid-reducing agents (ARAs) use among cancer patients of up to 33%, and PPIs were the most commonly prescribed [11]. In the literature, there were no known drug–drug interactions between anti-angiogenic monoclonal antibodies and PPIs [12]. However, in the molecular study, PPIs were reported to be able to induce VEGF expression and reduce the anti-tumor effects of bevacizumab [13]. These findings hinted that prolonged use of PPIs might result in long-term adverse events and potential drug–drug interactions, including anti-cancer treatment.

The gastric acid-reduction effects of histamine H2-receptor antagonists (H2RAs) are considerably lower than those of PPIs [14]. A previous molecular study found that histamine might upregulate VEGF and promote angiogenesis in the granulation tissue through the H2 receptor-cyclic AMP-PKA pathway [15]. Based on this finding, inhibition of histamine might be beneficial for anti-cancer treatment.

According to these molecular mechanisms, we hypothesize that PPIs might decrease the efficacy of bevacizumab, thereby negatively affecting the oncological outcomes of patients with CRC treated with bevacizumab. This retrospective cohort study was aimed to compare PPI users with H2RA users among patients with CRC, in terms of oncological outcomes, using a national claims database.

## 2. Materials and Methods

### 2.1. Ethics Statement

This study was approved by the Institutional Review Board (IRB) of the Dalin Tzu Chi Hospital (IRB approval number: B10704014) and was conducted in accordance with relevant guidelines and regulations. Prior to analysis, the Taiwan National Health Insurance (NHI) program removed all personally identifiable information from the initial dataset. De-identification ensures that no medical record can be traced to any individual patient. Therefore, informed consent could be waived under the approval of the IRB. The procedures used in this study adhere to the tenets of the Declaration of Helsinki.

### 2.2. Data Source

This dataset for research was obtained from the NHI Research Database (NHIRD), created by the Taiwan NHI. The data between 2005 and 2020 that were available for this study included all medical claims. Approximately 99% of the population of Taiwan and contracts with 97% of medical providers are covered by the Taiwan NHI, a single-payer insurance system. [16,17]. In addition to de-identified ID and date of visits, the medical information included diagnosis codes of diseases in ICD-9CM (before 2016) and ICD-10CM (since 2016) format, and therapy such as procedures and drugs in the form of anatomical therapeutic chemical codes. For the histological classification, the Taiwan Cancer Registry (TCR) was used to connect with the NHIRD dataset. The TCR is a nationwide population-based registry with a high degree of accuracy in TCR long-form data, which is provided by Ministry of Health and Welfare, Taiwan [18].

### 2.3. Study Population and PPI/H2RA Exposure

The study design is shown in Figure 1 and Figure 2. All bevacizumab prescription data between 2005 and 2020 was obtained from the National Health Insurance Research Database (N = 29,006). Patients with a diagnosis of CRC (*n* = 26,529) followed by at least two occurrences of bevacizumab prescriptions of duration >30 days were included (*n* = 20,778). The index date was defined as the date of the first bevacizumab prescription. After further excluding 21 patients miscoded as dead or dropped out of the insurance system before the index date, 20,757 patients remained. The study was designed in new users of acid-reducing agents (ARAs). The use of ARAs within 30 days after the index was calculated. One cohort was composed of PPI users and the other was composed of H2RA users. For dose–response relationship analysis, the cDDD of PPI were divided in 4 subgroups: 0–15, 15–32, 32–47, and ≥47. The patients without ARAs also compared with the PPI users and H2RA users, respectively; the study design flowchart is shown in Appendix A.

### 2.4. Confounding Factors and Propensity Match

Demographic variables were based on patients’ insurance enrollment records at the index date, including age, sex, urbanization, region, and monthly income The individual comorbidity, Charlson Comorbidity Index (CCI), and concomitant medication use were evaluated by the outpatient and inpatient medical claims in one year prior to the index date.

Patients who had more severe cancer status and comorbidities might have a higher consumption of analgesics, and thus more PPI/H2RA might be prescribed. This may be an indication bias while evaluating the relationship between PPI/H2RA use and prognosis of cancer treatment. Propensity score (PS) matching was used to decrease the effects of indication bias. We incorporated cancer stages, peptic ulcer disease, gastroesophageal reflux disease, gastrointestinal bleeding, steroid usage, and non-steroid anti-inflammatory drug (NSAIDs) usage into matching.

The propensity score was calculated from a logistic regression model based on age, sex, CCI, comorbidity, and concomitant medication use. Patients having the same metastasis cancer type, with the closest PS, were allowed to pair up if the difference between their PS was <0.0001. PS match was conducted for PPI users vs. H2RA users (1:1) separately, and post-matching groups became comparable.

### 2.5. Study Outcomes

The primary outcome was mortality, which was differentiated into all-cause mortality and cancer-specific mortality. All patients were followed up for this outcome from the index date until the end of year 2020 or censor.

### 2.6. Statistical Methods

Chi-squared tests were used for association in contingency tables while student t-tests were used for continuous variables. The empirical survival function was estimated using Kaplan–Meier methodologies, followed by the log-rank test and cumulative incidence of recurrence by CIF with Gray’s modified Chi-squared test. Adjusted hazard ratios were estimated using Cox’s proportional hazard model. All statistical analyses were performed using SAS version 9.4. The statistical analysis of the study was performed by a biostatistician.

## 3. Results

### 3.1. Baseline Comparability

Steroids (>99%) and non-steroidal anti-inflammatory drugs (>80%) were widely prescribed to this population. More patients had a history of peptic ulcer disease (PUD) and fewer with gastrointestinal (GI) bleeding. Urbanization, region, and socioeconomic status (monthly income) reflected the distribution of the general population. These variables were not included in the logistic regression model for PS calculation, but were included in the proportional hazard model to adjust for availability and quality of medical care among various groups in Taiwan (Table 1).

### 3.2. Incidence and Adjusted Hazard Ratio

The incidence proportion (IP) per 1000 person-year follow up of overall mortality and cancer-specific mortality was higher in PPI users than H2RA users, (464.1 and 451.6) than H2RA users (388.6 and 377.0 respectively). The results are shown in Table 2. To explore whether PPI and H2RA affect the overall survival of CRC patients treated with bevacizumab, the PPI users and H2RA users were matched with the patients without ARAs, and overall survival and cancer-specific survival were listed as Appendix A.

Cox modeling showed a significant risk of both overall mortality and cancer-specific mortality in PPI users, with a respective adjusted hazards ratio (aHR) of 1.19 (95% CI: 1.09, 1.31) and 1.20 (95% CI: 1.09,1.31) when compared to H2RAs users (Table 3).

### 3.3. Dose Response Relationship

A dose–response relationship was clearly observed in PPI users. The aHR of cDDD in the 0–15, 15–32, 32–47, and ≥47 subgroups were 1.02 (*p* = 0.7931), 1.17 (*p* = 0.0316), 1.30 (*p* = 0.0004), and 1.29 (*p* = 0.0005), respectively, for overall mortality versus H2RA users, and 1.03 (*p* = 0.7332), 1.18 (*p* = 0.0326), 1.29 (*p* = 0.0005), and 1.30 (*p* = 0.0005), respectively, for cancer-specific mortality versus H2RA users (Table 3).

Compared with non-users, both PPIs and H2RAs were associated with poorer overall survival and the effect was more significant in PPI users. (Appendix A) Compared with H2RAs, PPIs had a negative effect on overall survival in mCRC patients undergoing bevacizumab treatment (Figure 3).

## 4. Discussion

In this population-based matched retrospective cohort study, we found a different response for PPIs and H2RAs in mCRC patients undergoing bevacizumab treatment. Compared with ARA non-users, both PPI users and H2RA users were associated with poorer survival outcomes, while there was no significant dose-dependent manner in H2RA users. Compared with PPI users and H2RA users, PPI exposure was associated with an increased risk of overall death and cancer-specific death than patients using H2RAs. The risk increased with increasing PPI dose.

There was limited literature discussing the complex effects of different ARAs in cancer patients. In Yagi’s study, vonoprazan and PPIs demonstrated different effects on CRC and other cancer types. In CRC patients, PPIs might prolong the duration of bevacizumab, but the opposite may be true for other cancer types [19]. In mCRC patients taking PPIs, the anti-cancer effect of capecitabine might be compromised and treatment with fluoropyrimidine was more favorable in these patients [20]. However, there are no studies comparing the effects of PPIs and H2RAs in mCRC patients undergoing bevacizumab treatment. Our study showed that compared with the H2RAs, PPIs negatively affect survival in mCRC patients undergoing anti-VEGF treatment in a dose-dependent manner.

There are a few potential mechanisms that could lead to reduced therapeutic effects of anti-cancer agents in mCRC patients using PPIs. PPIs induced hypergastrinemia, which could stimulate carcinogenesis and cancer invasion [21]. PPIs have been reported to influence CRC cell line survival in vitro, including promoting cell growth and invasiveness [22] and VEGF expression transcription and translation [23]. Another possible mechanism is the drug–drug interactions (DDIs) with PPIs, which decreases drug absorption and negatively affects cancer treatment. This occurs through various mechanisms, such as altered microbiota induced by PPIs and their metabolism by cytochrome P450 enzymes [12]. Gut microbiota might interact with PPIs and the effect of the anti-cancer treatment. Dysbiosis caused by PPIs could lead to increased drug metabolism, altered autophagy, or immunosuppression [24].

There is also evidence that H2RA might positively affect the outcomes of CRC treatment. H2R has been reported to be potentially pro-carcinogenetic and H2RA might prolong the survival of CRC patients by blocking H2R. A meta-analysis has demonstrated that H2RAs might be effective for CRC adjuvant treatment and reduce overall CRC mortality, with a hazard ratio of 0.7 [25]. H2RAs might inhibit the angiogenetic activity through the reduction of VEGF and suppression of colon cancer growth in the mouse model [26]. H2RAs also improved immune response by promoting peri-tumoral lymphocyte growth and suppressing metastasis [27]. In this study, under the setting of mCRC patients undergoing anti-VEGF treatment, the PPIs were associated with a higher overall and cancer-related mortality than H2RAs.

### 4.1. Strengths of the Study

The major strengths of this study were the use of a nationwide population-based data and a large sample size. We performed a retrospective cohort study using a database for reimbursement of medical expenses as well as a national cancer registration system to evaluate the relationship between ARAs and oncological outcomes in mCRC patients undergoing bevacizumab treatment. Another strength of our study was that biases and covariates were considered. A person-years approach to determining incidence rate was performed to reduce bias from different points in time. In addition, we used propensity score matching to minimize the bias between PPI users and H2RA users. Finally, the cases and controls were collected under a new-user design and the observation period was well-designed for evaluating drug effects.

### 4.2. Limitations of the Study

This study had several limitations. First, patient adherence to medications could not by analyzed because of the prescription claims database in this study. For example, PPI exposure is measured by prescribed claims. Second, several potential risk behaviors such as dietary patterns, lifestyle, tobacco and alcohol consumption, and betel nut consumption were unavailable in the NIRHD database. Third, the most of study population was limited to Taiwan and might not be generalizable to other populations. Finally, this is a retrospective cohort study, rather than a randomized study.

## 5. Conclusions

The application of ARAs, such as PPI and H2Ras, in mCRC patients who underwent bevacizumab treatment contributes to the reduced overall survival. PPI use in mCRC patients undergoing bevacizumab treatment was associated with dose-dependent, poorer oncological outcomes than H2RA use. Along with paying more attention to the benefits and risks of long-term use of ARAs in mCRC patients with bevacizumab treatment, physicians should also reduce ARAs, especially PPI prescribing, to the extent possible for peptic ulcer and esophageal reflux.

## Figures and Tables

**Figure 1 cancers-16-03378-f001:**
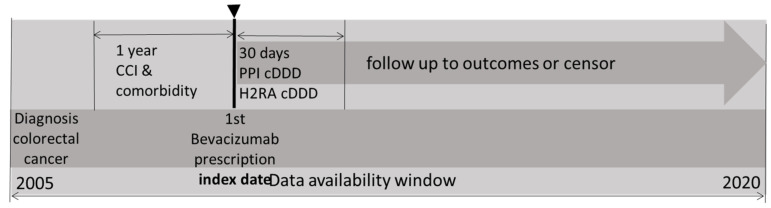
The diagram of study design. Patients with a diagnosis of CRC were included from 2005 to 2020. Patients with at least 2 prescriptions of bevacizumab were included and the index date was defined as the date of the first prescription of bevacizumab. The Charlson Comorbidity Index and comorbidities were evaluated 1 year prior to the index date. The use of PPIs and H2RAs within 30 days after the index date were calculated. The enrolled cases were followed up from the index date to the end of 2020 or until they were censored.

**Figure 2 cancers-16-03378-f002:**
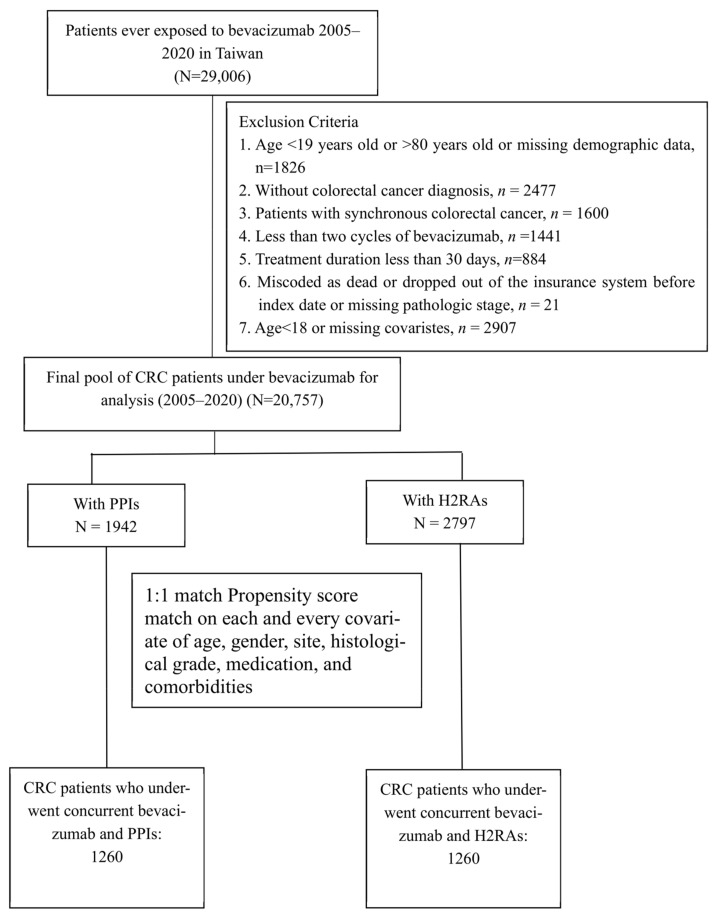
Study design flowchart of the cohort study. Data for CRC patients undergoing bevacizumab between 2005 and 2020 in Taiwan were obtained from the National Health Insurance Research Database and the Taiwan Cancer Registry. The comorbidities were evaluated based on the medical claims 1 year prior to the index date. All patients were followed up until 2020 or death, with at least a one-year follow-up period. CRC patients diagnosed between 2006 and 2019 were included.

**Figure 3 cancers-16-03378-f003:**
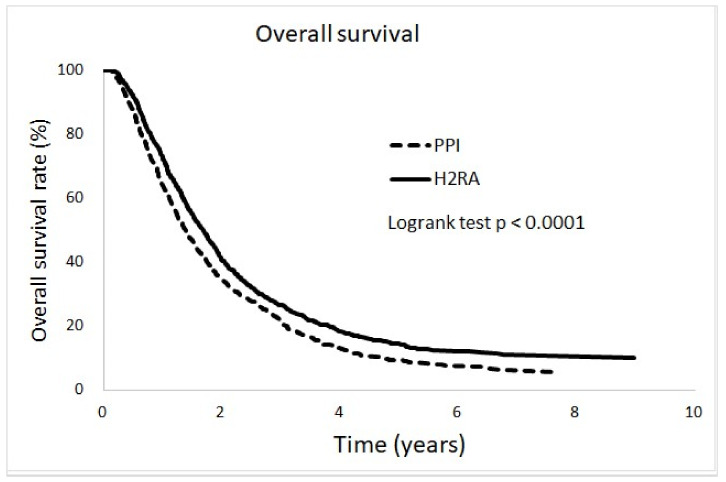
Proton pump inhibitors were associated with poorer overall survival than H2 receptor antagonists in mCRC patients. mCRC: metastatic colorectal cancer; PPI: proton pump inhibitors; H2RA: H2 receptor antagonist.

**Table 1 cancers-16-03378-t001:** Patients’ baseline characteristics after propensity score matching.

Covariate	PPI	H2RA	*p* Value
	N (%), *n* = 1260	N (%), *n* = 1260	
**Age (year)**			0.9981
20–54	290 (23.02)	292 (23.17)	
55–62	326 (25.87)	322 (25.56)	
63–69	312 (24.76)	312 (24.76)	
70–80	332 (26.35)	334 (26.51)	
**Sex**			1.000
Male	770 (61.11)	771 (61.19)	
Female	490 (38.89)	489 (38.81)	
**Cancer site**			1.000
Colon, left	379 (30.08)	379 (30.08)	
Colon, right	354 (28.10)	354 (28.10)	
Colon, unspecified	70 (5.56)	70 (5.56)	
Rectum	457 (36.27)	457 (36.27)	
**Pathological grade**			0.9871
1	19 (1.51)	19 (1.51)	
2	675 (53.57)	675 (53.57)	
3	77 (6.11)	76 (6.03)	
4 and others	489 (38.81)	490 (38.89)	
**Comorbidity**			
With PUD	386(30.63)	385 (30.56)	0.9655
With GI bleeding	31 (2.46)	28 (2.22)	0.6927
**Medication**			
NSAID usage	1054 (83.65)	1054 (83.65)	1.000
Steroid usage	1256 (99.68)	1260 (100)	0.1793
**CCI**			0.9998
<8	64 (5.08)	65 (5.16)	
8	461 (36.59)	461 (36.59)	
9	417 (33.10)	417 (33.10)	
>9	318(25.24)	317(25.16)	
**Urbanization**			0.9695
High	255 (20.24)	250 (19.84)	
Median	618 (49.05)	621 (49.29)	
Low	387 (30.71)	389 (30.87)	
**Region**			0.0048
North	452 (35.87)	459 (36.43)	
Central	360 (28.57)	287 (22.78)	
East	29 (2.30)	36 (2.86)	
South	419 (33.25)	478 (37.94)	
**SES (monthly income)**			0.2889
≤20.1 K	444 (35.24)	433 (34.37)	
20.1–22.8 K	205 (16.27)	218 (17.30)	
22.8–42 K	325 (25.79)	355 (28.17)	
≥42 K	286 (22.70)	254 (20.16)	

PPI: proton pump inhibitor; H2RA: H2 receptor antagonist; PUD: peptic ulcer disease; GI: gastrointestinal; NSAID: non-steroid anti-inflammation drug; CCI: Charlson comorbidity index; SES: socioeconomic status.

**Table 2 cancers-16-03378-t002:** Incidence proportion (IP) and event rate.

	All-Cause Death		CRC-Specific Death	
	IP (95% CI)	Event (%)	IP (95% CI)	Event (%)
**PPIs vs. H2RAs**				
PPIs	464.1 (434.8, 495.0)	928 (73.7)	451.6 (422.7, 482.1)	903 (71.7)
H2RAs	388.6 (363.2, 415.3)	870 (69.0)	377.0 (352.0, 403.3)	844 (67.0)

CRC: colorectal cancer; PPI: proton pump inhibitor; H2RA: H2 receptor antagonist; IP: incidence proportion per 1000 person-years.

**Table 3 cancers-16-03378-t003:** The dose-dependent effect of antacids and overall survival and cancer-specific death.

	All-Cause Death		Cancer-Specific Death	
	Adjusted HR (95% CI)	*p* Value	Adjusted HR (95% CI)	*p* Value
**PPIs vs. H2RAs**	1.19 (1.09,1.31)	0.0002	1.20 (1.09, 1.31)	0.0002
**Dose Response**				
cDDD 0–15	1.02 (0.87, 1.19)	0.7931	1.03 (0.88, 1.20)	0.7332
cDDD 15–32	1.17 (1.01, 1.36)	0.0316	1.18 (1.01, 1.36)	0.0326
cDDD 32–47	1.30 (1.12, 1.49)	0.0004	1.29 (1.12, 1.49)	0.0005
cDDD ≥ 47	1.29 (1.12, 1.49)	0.0005	1.30 (1.12, 1.50)	0.0005

HR: hazard ratio; CI: confidence interval; PPI: proton pump inhibitor; H2RA: H2 receptor antagonist; cDDD: cumulative defined daily dose.

## Data Availability

According to the policy of the Health and Welfare Data Science Center, Ministry of Health and Welfare, Taiwan, the datasets generated and/or analyzed during the current study are not publicly available. The datasets are available from the corresponding author upon reasonable request.

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
