# Peer review of "Proton Pump Inhibitors Worsen Colorectal Cancer Outcomes in Patients Treated with Bevacizumab"

_cancers, 2024, doi:10.3390/cancers16193378_

Round 1

Reviewer 1 Report

Comments and Suggestions for Authors

The authors of the manuscript investigated the clinical survival of metastatic colon cancer patients prescribed with either PPI and bevacizumab, and H2RA and bevacizumab. The study utilized large patient cohorts and adopted a logistic regression model to calculate propensity scores for comparison between PPI and H2RA patients. The results showed that PPI patients have lower overall and cancer-specific survival, and associated with poorer outcome in a dose-dependent manner. 

It is logical that the authors adopted such a methodology to remove bias in comparison, what does overall survival look like for all patients that satisfy selection criteria (n = 1942, n = 2797, respectively)?

Comments on the Quality of English Language

Good presentation and writing. Some minor typos, such as in line 44. 

Reviewer 2 Report

Comments and Suggestions for Authors

The effects of proton pump inhibitors (PPIs) and H2 receptor antagonists (H2RAs) on the survival of patients of metastasis colorectal cancer treated with bevacizumab were analysed and compared. It has been demonstrated that PPI application was associated with dose-dependent poorer oncological outcomes in metastatic colorectal cancer (CRC) patients treated with bevacizumab when compared to H2RA.

It is also critical to show the results from the comparison of PPI users to non-PPI users or non-acid-reducing agent users within the patient cohort treated with bevacizumab to determine if PPI users have lower survival rates overall to conclude that proton pump inhibitors worsen the outcomes of CRC patients treated with bevacizumab.

From the results presented, it can only be concluded that compared to H2RA, PPI reduced the survival of colorectal cancer patients treated with bevacizumab. Therefore, H2RAs may be selected over PPI when prescribing anti-acid agents to CRC patients.  It is unclear whether PPI and H2RA affect the overall survival of CRC patients treated with bevacizumab that will help make justification when considering the application of anti-acid agents to CRC patients.

Round 2

Reviewer 2 Report

Comments and Suggestions for Authors

The conclusion should be justified to reflect the results that the application of ARAs, such as PPI and H2RAs in patients of mCRC contributes to the reduced overall survival of the disease.

Comments on the Quality of English Language

Minor editing of the English language is required.
